# Fish larval recruitment to reefs is a thyroid hormone-mediated metamorphosis sensitive to the pesticide chlorpyrifos

Guillaume Holzer[1†*], Marc Besson[1,2,3†*], Anne Lambert[1], Loïc François[2], Paul Barth[1], Benjamin Gillet[1], Sandrine Hughes[1], Gwenaël Piganeau[3], Francois Leulier[1], Laurent Viriot[1], David Lecchini[2,4‡], Vincent Laudet[1‡*]

[1]Institut de Génomique Fonctionnelle de Lyon, Université de Lyon, Université Claude Bernard Lyon 1, UMR CNRS 5242, Ecole Normale Supérieure de Lyon, Lyon, France; [2]CRIOBE USR3278 EPHE-UPVD-CNRS, PSL Research University, Moorea, French Polynesia; [3]Observatoire Océanologique de Banyuls-sur-Mer, UMR CNRS 7232, Université Pierre et Marie Curie Paris, Paris, France; [4]Laboratoire d'Excellence CORAIL, Moorea, French Polynesia

*For correspondence:
guillaume.holzer@ens-lyon.fr (GH);
marc.besson@ens-lyon.org (MB);
vincent.laudet@obs-banyuls.fr (VL)

[†]These authors contributed equally to this work
[‡]These authors also contributed equally to this work

Competing interests: The authors declare that no competing interests exist.

**Abstract** Larval recruitment, the transition of pelagic larvae into reef-associated juveniles, is a critical step for the resilience of marine fish populations but its molecular control is unknown. Here, we investigate whether thyroid-hormones (TH) and their receptors (TR) coordinate the larval recruitment of the coral-reef-fish *Acanthurus triostegus*. We demonstrate an increase of TH-levels and *TR*-expressions in pelagic-larvae, followed by a decrease in recruiting juveniles. We generalize these observations in four other coral reef-fish species. Treatments with TH or TR-antagonist, as well as relocation to the open-ocean, disturb *A. triostegus* larvae transformation and grazing activity. Likewise, chlorpyrifos, a pesticide often encountered in coral-reefs, impairs *A. triostegus* TH-levels, transformation, and grazing activity, hence diminishing this herbivore's ability to control the spread of reef-algae. Larval recruitment therefore corresponds to a TH-controlled metamorphosis, sensitive to endocrine disruption. This provides a framework to understand how larval recruitment, critical to reef-ecosystems maintenance, is altered by anthropogenic stressors.
DOI: https://doi.org/10.7554/eLife.27595.001

## Introduction

Life history transitions are critical for many animal species and often correspond to concomitant developmental and ecological shifts (*Bishop et al., 2006*). Unfortunately, little is known about how internal and external cues act in concert during these events. The life cycle of most teleost reef fish include a major developmental and ecological transition. Adults reproduce in the vicinity of the reef, emitting eggs that disperse and hatch in the ocean, where the larvae grow (*Leis and McCormick, 2002*). Larvae migrate back and enter reefs where they become juveniles, a step called larval recruitment (*Leis and McCormick, 2002*). This step involves the perception of environmental cues for larvae to localize and settle in the reef, and is accompanied by major morphological changes (*McCormick et al., 2002*; *Lecchini et al., 2005b*, *2005a*; *Lecchini et al., 2013*; *Barth, 2015*). This transition of pelagic larvae into reef-associated juveniles is often referred to as metamorphosis and is critical for the maintenance of reef fish populations, but its molecular control remains largely unknown (*Dufour and Galzin, 1993*; *Doherty, 2002*; *Leis and McCormick, 2002*; *McCormick et al., 2002*; *Dixson et al., 2011*; *Barth, 2015*). Since larval recruitment is an ecological

**eLife digest** Many animals go through a larval phase before developing into an adult. This transformation is called metamorphosis, and it is regulated by hormones of the thyroid gland in vertebrates. For example, most fish found on coral reefs actually spend the first part of their life as free-swimming larvae out in the ocean. The larvae usually look very different from the juveniles and adults. When these fish become juveniles, the larvae undergo a range of physical and behavioral changes to prepare for their life on the reef. Yet, until now it was not known what hormones control metamorphosis in these fish.

To address this question, Holzer, Besson et al. studied the convict surgeonfish *Acanthurus triostegus*. This herbivorous coral-reef fish lives in the Indo-Pacific Ocean, and the results showed that thyroid hormones do indeed regulate the metamorphosis of its larvae. This includes changing how the larvae behave and how their adult features develop. Further, Holzer, Besson et al. found that this was also true for four other coral-reef fish, including the lagoon triggerfish and the raccoon butterflyfish. In *A. triostegus*, thyroid hormones controlled the changes that enabled the juveniles to efficiently graze on algae growing on the reef such as an elongated gut.

When the fish larvae were then exposed to a pesticide called chlorpyrifos, a well-known reef pollutant, their hormone production was disturbed. This in turn affected their grazing behavior and also their metamorphosis. These fish had shortened, underdeveloped guts and could not graze on algae as effectively.

Herbivorous fish such as *A. triostegus* play a major role in supporting coral reef ecosystems by reducing algal cover and therefore promoting coral recruitment. These new findings show that pollutants from human activities could disturb the metamorphosis of coral-reef fish and, as a consequence, their ability to maintain the reefs. A next step will be to test what other factors can disrupt the hormones in coral-reef fish and thus pose a threat for fish populations and the coral-reef ecosystem.

DOI: https://doi.org/10.7554/eLife.27595.002

event coupled to a morphological transformation, reminiscent of the situation in amphibians, it may correspond to a thyroid hormones (TH) controlled metamorphosis (*Bishop et al., 2006*; *Brown and Cai, 2007*).

TH and their receptors (TR) trigger and coordinate metamorphosis of many species such as *Xenopus* (*Brown and Cai, 2007*) or flatfish (*Solbakken et al., 1999*). The thyroid gland produces mainly the thyroxine hormone (T4), which is peripherally transformed into triiodothyronine (T3), the active form (*Chopra, 1996*). The transformation of T4 into T3, and the degradation of T4 and T3 are controlled by a family of enzymes called deiodinases (*Bianco and Kim, 2006*). TH levels peak at the climax of metamorphosis. Precocious treatment with TH triggers metamorphosis whereas goitrogen (TH synthesis inhibitors) treatment blocks it (*Tata, 2006*). Similar mechanisms have been described in several teleost fishes, but in contexts disconnected from the natural environment (*Brown, 1997*; *de Jesus et al., 1998*; *Yamano and Miwa, 1998*; *Kawakami et al., 2003*; *Marchand et al., 2004*; *McMenamin and Parichy, 2013*).

In the convict surgeonfish *Acanthurus triostegus,* larvae have a planktonic diet while juveniles and adults are herbivorous, and external morphological changes occurring at recruitment have been previously described (*Randall, 1961*; *McCormick, 1999*; *McCormick et al., 2002*; *Frédérich et al., 2012*). Interestingly, crest-captured larvae can delay their metamorphosis when relocated to the external slope of the reef (open ocean), showing an influence of the environment on their morphological transformation (*McCormick, 1999*). Nevertheless, again, the molecular mechanisms remain unknown.

Here we report that *A. triostegus* larval recruitment to the reefs corresponds to metamorphosis with major pigmentation changes, remodeling of the digestive tract and behavioral modifications. Our results show that TH levels and *TR* expressions control this remodeling process. Furthermore, recruiting larvae can freeze their TH signaling when relocated on the external slope, demonstrating an influence of the environment on metamorphosis. Chlorpyrifos (CPF), a known reef pollutant and endocrine disruptor (*Roche et al., 2011*; *Botté et al., 2012*; *Juberg et al., 2013*; *Slotkin et al.,*

*2013*), affects *A. triostegus* metamorphosis by impairing TH signaling, preventing intestine lengthening, and inhibiting fish grazing activity. CPF therefore impairs *A. triostegus* control on algae spreading, which is a major threat for reef conservation. We extend our observation upon TH levels to four other coral reef fish species. Our work provides a unifying framework that integrates the developmental, ecological and evolutionary perspectives of vertebrate life history transitions. The involvement of the TH signaling pathway in this key post-embryonic step, thus prone to endocrine disruption, provides an obvious entry point to study how anthropogenic stressors may affect reef fish populations.

## Results

### *Acanthurus triostegus* larvae undergo a major and rapid remodeling at recruitment

*A. triostegus* crest captured individuals (n = 297) experience an important weight loss between day 1 (0.71 ± 0.01 g) and day 8 (0.53 ± 0.01 g) following their entry in the reef (*Figure 1A–B*). Individuals also undergo extensive pigmentation changes (*Figure 1C–L*), as exemplified by the very rapid appearance of black stripes (less than four hours, MB pers. obs.) after entering the reef (*Figure 1C–D, H-I*). This is followed by the widening of these vertical black stripes and by the onset of the body's white pigmentation (*Figure 1D–G*, *Figure 1I–L*).

Furthermore, using conventional X-Ray microtomography, we identified three different dental generations in crest larvae to day 8 juveniles (red, blue and green, *Figure 1M–Q*). Crest larvae present tiny and poorly mineralized pointed teeth at the distal ends of the jaws (red, *Figure 1M*, *Figure 1—figure supplement 1R*) that are likely remnants of an oceanic larval dentition (dentition A). Other medium-sized and more mineralized teeth are in function in medial areas of the jaws (blue, *Figure 1M*). These teeth display pointed cusps and belong to a more advanced dental generation (dentition B). Lastly, much larger teeth of the next generation (green, dentition C) are about to erupt from jaws in crest larvae (*Figure 1M*) and replace A and B previous dentitions in day 2 to day 8 juveniles (*Figure 1N–Q*). These teeth are highly similar to the shovel-shaped adult teeth (*Figure 1—figure supplement 1S*) that serve for grazing algae on hard substrata. The rapid formation of dentition C is consistent with a diet shift from planktivorous to herbivorous at recruitment. This shift is precisely organized, since it begins in the sagittal area of all individuals, and secondarily extends to the distal parts of the jaw (*Figure 1M–Q*). The fact that crest captured larvae already have teeth of the C dentition about to erupt (*Figure 1M*) indicates that fish are prepared for reef life conditions before reef entry.

The intestine also goes through a drastic remodeling at recruitment. In particular, the gastrointestinal tract lengthens from about 2.52 ± 0.07 cm in crest captured larvae to 7.38 ± 0.14 cm in day 8 juveniles (*Figure 1W*). To characterize this change, we performed histological sections on the middle part of the intestine during the larval recruitment process (*Figure 1R–V*). Intestines of crest larvae and day 2 juveniles have a thick wall with many muscle fibers and regular villi (*Figure 1R and S*). At day 3, intestines undergo a spectacular remodeling with the disappearance of the epithelium, leaving only the muscular layer (*Figure 1T*), while epithelium reformation occurs later at day 5 (*Figure 1U*). In day 8 individuals, the intestine is fully remodeled: the epithelium is thicker and exhibits villi again (*Figure 1V*). Such spectacular remodeling events are also observed in the proximal and distal portion of the intestines (*Figure 1—figure supplement 1A–O*).

To assess whether and to what extent this intestine remodeling was correlated to a diet shift at larval recruitment, we sequenced *A. triostegus* digestive tract content (18S mass sequencing) to identify its eukaryotic composition. We investigated crest captured individuals, day 8 juveniles and adults (*Figure 1W*) using a blocking primer to minimize amplification of host sequences. Crest larval intestines are largely empty although we identified sequences of fungi and algae. In day 8 juveniles, the intestinal content is different with more brown and red algae, as well as few protostome sequences. This indicates a shift in the eukaryotic gut content between crest individuals and day 8 juveniles. Adults gut content is less diversified and is mainly composed of brown algae, red algae and alveolates, different from those found in day 8 juveniles (*Figure 1W*). Using 16S metagenomic sequencing, we found differences in the relative quantity of bacterial community between the juveniles at day 2, 5 and 8 (*Figure 1—figure supplement 1P*). The 16S profile of the adult is also different from

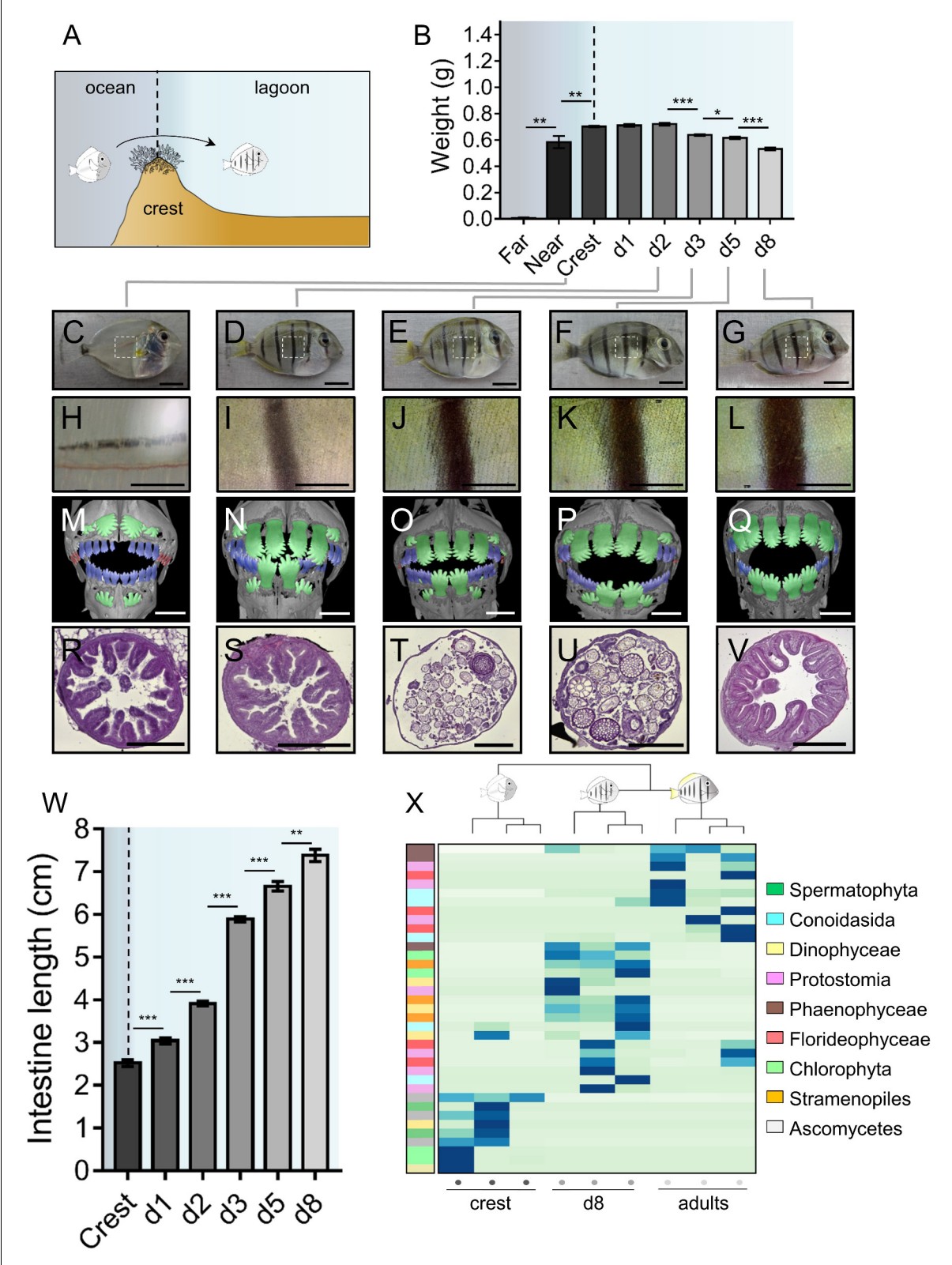

**Figure 1.** Changes in *Acanthrurus triostegus* during larval recruitment. Error bars represent standard errors. *p-value<0.05; **p-value<0.01; ***p-value<0.001. (**A**) Schematization of the reef colonization by a larva (left) turning into a juvenile (right). The background color shading centered into the reef crest is re-used in following figures. (**B**) Weight variation during larval recruitment. n = 3 biological replicates for 5-to-10 km offshore fish (far), n = 5 for 2 km offshore fish (near) and n = 297 for crest to day 8 (d8) fish. Statistics: mean comparisons (Wilcoxon-Mann-Whitney U-test). See *Figure 1—*

*Figure 1 continued on next page*

*Figure 1 continued*

*source data 1* for detailed data and statistics. (C–G) Variation in pigmentation and morphology at different sampling points, (C) crest larva, (D) day 2 juvenile, (E) day 3 juvenile, (F) day 5 juvenile and (G) day 8 juvenile; scale bars indicate 0.5 cm. The white square represent the magnified region shown in the line below. These pictures are representative of n > 50 fish (biological replicates) per developmental stage, sampled at n > 20 different times during the year (technical replicates). (H–L) Close-up on the pigmented middle body band. Scale bars indicate 0.2 cm. These pictures are representative of n > 50 fish (biological replicates) per developmental stage, sampled at n > 20 different times during the year (technical replicates). (M–Q) X-Ray microtomography of larva, dentition A, B or C are highlighted respectively in red, blue or green. Scale bars indicate 0.5 mm. These pictures are representative of n = 2 fish (biological replicates) per developmental stage. (R–V) Histological sections (5 µm width) of intestines with haematoxylin/eosin staining. Scale bars indicate 200 µm. These pictures are representative of n = 6 sectioning technical replicates, performed twice (sampling technical replicates), on n = 2 fish (biological replicate). (W) Intestine length variation of *A. triostegus* during its larval recruitment. n = 9 to 14 biological replicates for each developmental stage (from n = 2 technical sampling replicates). Statistics: mean comparisons (Wilcoxon-Mann-Whitney U-test). See *Figure 1—source data 2* for detailed data and statistics. (X) Heatmap clustering of the 18S sequences identified in the intestine content of crest individual (n = 3, left panel), day 8 juveniles (n = 3, middle panel) and adults (n = 3, right panel). Each panel is made of three columns corresponding to one individual. Each line corresponds to one taxonomic group found in the intestines: *Spermatophyta* (dark green), *Conoidasisa* (cyan), *Dinophyceae* (light yellow), *Protostomia* (light mink), *Phaeophyceae* (brown), *Florideophycae* (dark pink), *Chlorophyta* (light green), *Stramenopiles* (orange) and *Ascomycetes* (grey).

DOI: https://doi.org/10.7554/eLife.27595.003

The following source data and figure supplement are available for figure 1:

**Source data 1.** *Acanthurus triostegus* weight at different developmental stages.
DOI: https://doi.org/10.7554/eLife.27595.005

**Source data 2.** *Acanthurus triostegus* intestine length at different developmental stages.
DOI: https://doi.org/10.7554/eLife.27595.006

**Figure supplement 1.** Changes in *Acanthurus triostegus* intestine and teeth during larval recruitment.
DOI: https://doi.org/10.7554/eLife.27595.004

the juvenile profile, with the notable presence of *Lachnospiraceae*, among which the giant bacteria *Epulopiscium fishelsoni* is a well-known symbiont of Acanthuridae (*Clements and Bullivant, 1991*).

## Thyroid hormone signaling peaks at larval recruitment

Given the role of TH in vertebrate metamorphosis (*Laudet, 2011*) and the spectacular transformations that *A. triostegus* undergoes at larval recruitment, we analyzed TH levels in larvae captured in the far ocean (from 5 to 10 km offshore, n = 3), near ocean (2 km offshore, n = 5), crest-captured larvae and juveniles (raised in situ in lagoon cages, n = 297) up to day 8 after recruitment (*Figure 2A–B*).

T4 levels are low in far ocean larvae and rise in fish located closer to the reef (near ocean larvae), before dropping 6-fold between the near ocean larvae and the day 8 juveniles (p-value=0.001, *Figure 2A*). This indicates a peak of T4 in near ocean and in crest larvae. Far ocean larvae exhibit a high level of T3 but with an important variability (*Figure 2B*). Although these larvae live in the same ecological niche (*i.e.* open ocean) and are roughly of the same size (4.7, 5.5, and 6.3 mm), they were captured in different locations and are probably not at the same developmental stage. Near ocean and crest captured larvae have similar levels of T3 that are 6-fold higher than the level observed in day 8 juveniles (p-value < 0.001 *Figure 2B*).

We characterized the three TR of *A. triostegus*: TRα-A, TRα-B and TRβ. These receptors have between 86% (TRα-A) and 96% (TRβ) sequence identity with their zebrafish orthologs (*Figure 2—figure supplement 1A*). The phylogenetic analysis indicates that each of the three TR clusters with its respective teleosts orthologs (*Figure 2—figure supplement 1B*). Moreover, transactivation assays show that these receptors behave as genuine TR (*Figure 2—figure supplement 1C*). We thus assessed the expression levels of *TRα-A*, *TRα-B*, and *TRβ* during *A. triostegus* development (*Figure 2C*), as well as *Klf9*, a conserved TR regulated gene (*Denver and Williamson, 2009*). Far ocean larvae have two profiles: one with a very low *TR* and *Klf9* expressions ('Far a' ocean) and one with very high levels of expression ('Far b' ocean) (*Figure 2C*). This suggests that there are, at least, two distinct situations in far ocean larvae: larvae with low TH/*TR* levels and larvae in which TH/*TR* levels surge. Far b ocean, near ocean, and crest larvae have high expression levels of the all the *TR* as well as *Klf9*, indicating a strong activation of the TH signaling at these stages. We observed a drop of all *TR* expression between crest larvae and day 8 juveniles (p-values=0.001, 0.005 and 0.002 for *TRα-A*, *TRα-B* and *TRβ* respectively, *Figure 2C*).

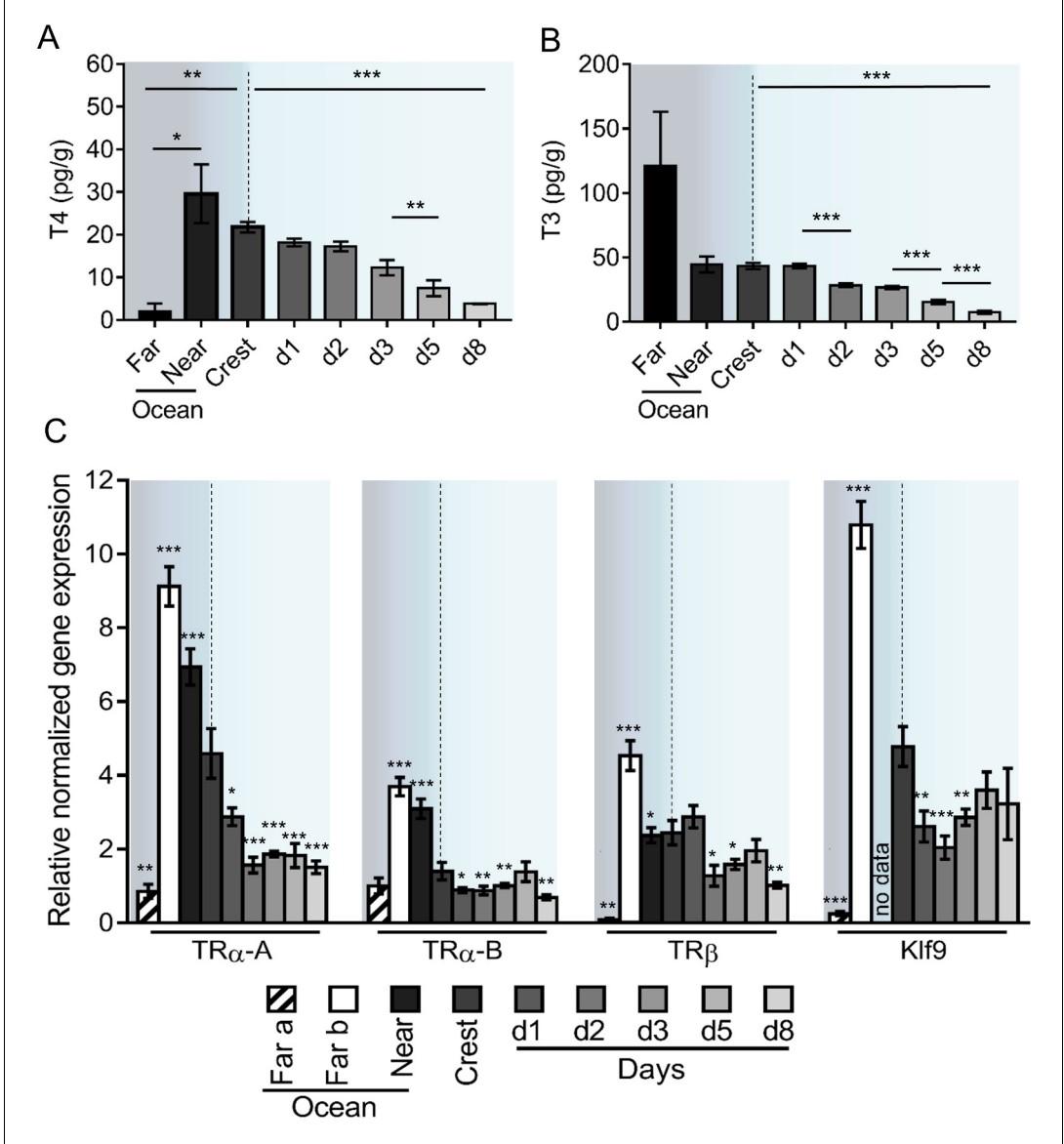

**Figure 2.** Thyroid hormone signaling during *Acanthurus triostegus* metamorphosis. Error bars represent standard errors. *p-value<0.05; **p-value<0.01; ***p-value<0.001. (A) T4 and (B) T3 dosage: Y-axis: T4 and T3 levels in pg.g$^{-1}$; X-axis: sampling point. n = 3 for 5 to 10 km offshore fish (far), n = 5 for 2 km offshore fish (near), n = 297 for crest to day 8 (d8) fish. Statistics: mean comparisons (Wilcoxon-Mann-Whitney U-test). See *Figure 2—source data 1* and *Figure 2—source data 2* for detailed data and statistics. (C) Expression level of *TRα-A, TRα-B, TRβ* and *Klf9* by qPCR, normalized with *Pold2* and *Rpl7*. Y-axis; Relative normalized expression X-axis: gene and sampling point.. Far ocean larvae are split in two groups: Far a (black stripes) and Far b (white). For each panel, the color shading is the same one as in *Figure 1A*. For each sampling point three biological replicates and three technical replicates were performed. Statistics: comparisons of each mean to the Crest condition mean (Student's t cumulative distribution function, which was automatically computed by qPCR software CFX Biorad Manager, *CFX Manager SH, 2017*). See *Figure 2—source data 3* for detailed data and statistics.

DOI: https://doi.org/10.7554/eLife.27595.007

The following source data and figure supplements are available for figure 2:

**Source data 1.** *Acanthurus triostegus* T4 levels in larvae and juveniles.
DOI: https://doi.org/10.7554/eLife.27595.011

**Source data 2.** *Acanthurus triostegus* T3 levels in larvae and juveniles.
DOI: https://doi.org/10.7554/eLife.27595.012

**Source data 3.** *TR* and *Klf9* expression levels in *Acanthurus triostegus* at different developmental stages (qPCR).
DOI: https://doi.org/10.7554/eLife.27595.013

**Source data 4.** Transactivation assay of *Acanthurus triostegus TR* for several thyroid hormone derivatives.

*Figure 2 continued on next page*

*Figure 2 continued*

DOI: https://doi.org/10.7554/eLife.27595.014

**Source data 5.** Transactivation assay of *Acanthurus triostegus TR* for the competition of T3 and NH3.

DOI: https://doi.org/10.7554/eLife.27595.015

**Source data 6.** List of primers used in this study.

DOI: https://doi.org/10.7554/eLife.27595.016

**Source data 7.** List of *TR* used for phylogenetic reconstruction.

DOI: https://doi.org/10.7554/eLife.27595.017

**Figure supplement 1.** Characterization of *Acanthurus triostegus* thyroid hormone receptors.

DOI: https://doi.org/10.7554/eLife.27595.008

**Figure supplement 2.** Primers used in this study.

DOI: https://doi.org/10.7554/eLife.27595.009

**Figure supplement 3.** List of thyroid hormone receptors sequences used for the phylogeny.

DOI: https://doi.org/10.7554/eLife.27595.010

## Thyroid hormone signaling and the environment control larvae transformation

To link the TH pathway, the environment and the metamorphic changes occurring at larval recruitment, we disturbed the TH signaling and the location of crest-captured larvae during their metamorphosis (from capture to up to day 8 after reef colonization).

In order to study how the environment controls the metamorphosis processes, we first relocated crest-captured larvae back to the external slope immediately after their reef entry. Indeed, mimicking oceanic condition has been acknowledged to delay metamorphosis (*McCormick, 1999*) (*Figure 3A*, upper panel). Similar to this earlier study, we observed a striking delay in the white pigmentation appearance between the vertical black stripes in day 2 relocated fish compared to their lagoon raised counterparts (*Figure 3A*, lower panel). The levels of T4 and T3 in fish relocated on the external slope at day 2, day 5 and day 8 are overall higher than the levels of their control counterparts in the lagoon (*Figure 3B–C*). The levels of T4, in particular, remain similar in day 2 to day 8 juveniles to the levels in the crest condition (*Figure 3B*). The expressions of *TRs* and *Klf9* in the relocated juveniles were also higher than in control juveniles at the same age (*Figure 3E*). This suggests that the environment effectively controls the metamorphosis by modulating the TH signaling.

To assess the implication of the TH signaling pathway on metamorphic processes, we then raised crest-captured larvae in situ in the lagoon and injected them daily, in their ventral cavity, with 20 μl of different drug solutions that activate or disrupt their TH molecular pathway. Four drug treatments were applied: (i) solvent control (DMSO diluted 10.000 times in Phosphate Buffer Saline 1X, as all drugs were made soluble in DMSO and diluted 10.000 times in PBS 1X); (ii) T3 +iopanoic acid (IOP) both at $10^{-6}$ M, IOP being used as an inhibitor of the deiodinase enzymes that therefore prevents the degradation of injected T3 (*Galton, 1989*; *Little et al., 2013*; *Lorgen et al., 2015*); (iii) NH3 at $10^{-6}$ M, NH3 being used as an antagonist of TR that prevents the binding of TH on TR (*Lim et al., 2002*; *Renko et al., 2012*) therefore disrupting *A. triostegus* TH pathway (*Figure 2—figure supplement 1D*); and (iv) T3 + IOP + NH3 all at $10^{-6}$ M to ensure the non-toxicity of NH3.

We used intestine total length, internal structure of guts (intestinal microvilli), dentition and pigmentation as markers of the advancement of metamorphosis since they strongly change after reef recruitment (*Figure 1*). T3 + IOP treatment increases the length of the intestine in day 2 and day 5 juveniles (p-values<0.001, *Figure 3E*, blue). On the contrary, NH3 treatment partially prevents the intestines lengthening occurring after recruitment (p-values<0.001, *Figure 3E*, green). Injections of T3 + IOP + NH3 result in an intestine length similar to the control individuals at day 2 and day 5 (p-values=0.486 and 0.444 respectively, *Figure 3E*). This shows that T3 and NH3 effectively compete, and confirms the non-toxicity of NH3. In parallel, intestines of fish relocated on the external slope are shorter than those of their lagoon counterparts at day 2 and day 5 (p-value<0.001 and p-value=0.042 respectively, *Figure 3E*, orange). This shows that their development has been slowed down, similarly to what we observed with the NH3 treatment. A similar pattern was observed concerning the microvilli remodeling within guts. In the anterior part of the guts, while T3 + IOP treatment accelerates the loss of the intestinal microvilli at day 2 and accelerates the development and the thickening of a new epithelium at day 5 (*Figure 3F*, second column), NH3 treatment and external

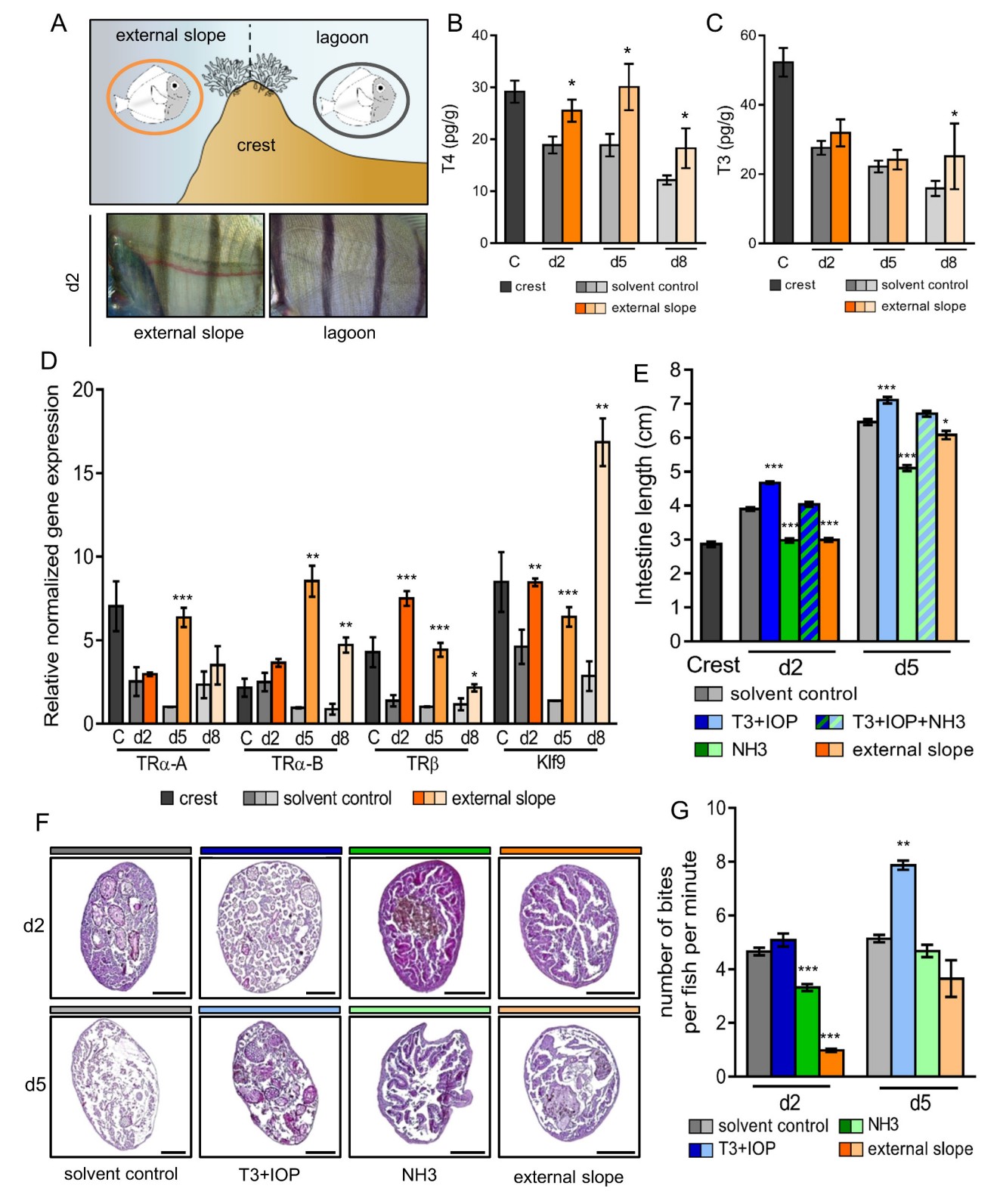

**Figure 3.** Impact of the environment and thyroid hormone signaling on *Acanthurus triostegus* metamorphic processes. Error bars represent standard errors. *p-value<0.05; **p-value<0.01; ***p-value<0.001. (**A**) Upper panel: schematization of the external slope experiment with control fish, in the lagoon (grey circle), and relocated fish, on the external slope (orange circle). Lower panel: comparison of the pigmentation between the relocated fish (left) and the control lagoon fish (right) at day 2. (**B**) T4 and (**C**) T3 dosage of crest captured larvae and day 2, 5 and 8 juveniles, in control (grey shade)

*Figure 3 continued on next page*

eLIFE Research article

Developmental Biology and Stem Cells | Ecology

*Figure 3 continued*

or external slope (orange shade) conditions. n = 4 fish for each experimental condition. X-axis: sampling point, Y-axis: T4 or T3 level in pg.g$^{-1}$. Statistics: mean comparisons between control and external slope (Wilcoxon-Mann-Whitney U-test). See *Figure 3—source data 1* and *Figure 3—source data 2* for detailed data and statistics. (D) Expression level of *TRα-A, TRα-B, TRβ* and *Klf9* by qPCR, normalized with *pold2* and rpl7 housekeeping genes, in control (grey shade) or external slope (orange shade) conditions. Y-axis: Relative normalized expression. X-axis: gene and stages. For each point three biological replicates and three technical replicates were performed. Statistics: comparisons of 'external slope' means to their 'control' counterparts (Student's t cumulative distribution function). See *Figure 3—source data 3* for detailed data and statistics. (E) Intestine length in cm of fish at crest (C), day 2 (d2) or day 5 (d5) under different treatment conditions: solvent-control (grey shade), T3 +IOP treated at 10$^{-6}$ M (blue shade), NH3 treated at 10$^{-6}$ M (green shade), T3 +IOP + NH3 treated at 10$^{-6}$ M (blue and green dashed shade) or located to the external slope (orange shade). n > 10 fish for each experimental condition. Statistics: ANOVA followed by Tukey posthoc tests, only significant differences with the respective solvent-control condition are indicated. See *Figure 3—source data 4* for detailed data and statistics (F) Histological cross section (5 µm width) of intestine stained by haematoxylin/eosin. Scale bars indicate 200 µm. These pictures are representative of n = 2 microtome section and staining technical replicates, performed on n = 3 fish (biological replicate) per treatment and per developmental stage. Two technical replicates of this experiment were performed. (G) Mean number of bites (on turf algae) per fish in a batch of 18 fish at day 2 (d2) and day 5 (d5) under different treatment conditions: solvent control (DMSO, grey shade), T3 + IOP at 10$^{-6}$µM (blue shade), NH3 at 10$^{-6}$µM (green shade) or relocated to the external slope (orange shade). n = 3 to 6 independent bite counts (technical replicates) for each condition. Bite counting was not performed on crest captured individuals as *A. triostegus* larvae do not feed for at least 12 hr after colonizing the reef at night (MB pers. obs.). Statistics: same as in (E). See *Figure 3—source data 5* for detailed data and statistics.

DOI: https://doi.org/10.7554/eLife.27595.018

The following source data and figure supplements are available for figure 3:

**Source data 1.** T4 levels in external slope relocated *Acanthus triostegus.*
DOI: https://doi.org/10.7554/eLife.27595.021

**Source data 2.** T3 levels in external slope relocated *Acanthus triostegus.*
DOI: https://doi.org/10.7554/eLife.27595.022

**Source data 3.** *TR* and *Klf9* expression levels in external slope relocated *Acanthurus triostegus* (qPCR).
DOI: https://doi.org/10.7554/eLife.27595.023

**Source data 4.** *Acanthurus triostegus* intestine length under pharmacological treatments or relocation.
DOI: https://doi.org/10.7554/eLife.27595.024

**Source data 5.** *Acanthurus triostegus* grazing activity under pharmacological treatments or relocation.
DOI: https://doi.org/10.7554/eLife.27595.025

**Figure supplement 1.** Thyroid hormone signaling and environment impact on the dentition and guts histology of *Acanthurus triostegus* at metamorphosis.
DOI: https://doi.org/10.7554/eLife.27595.019

**Figure supplement 2.** Pigmentation patterns and thyroid hormone signaling in *Acanthurus triostegus* at day 2 (d2) and day (d5) during larval recruitment.
DOI: https://doi.org/10.7554/eLife.27595.020

slope relocation prevent microvilli renewal at both day 2 and day 5 (*Figure 3F*, third and fourth column). These observations can be extended to the anterior, medium and posterior sections of the guts (see *Figure 3—figure supplement 1*). These changes are consistent with the notion that TH control remodeling of the guts at metamorphosis in *A. triostegus*. We did not observe any effect on teeth development neither with T3 + IOP or NH3 treatment, nor with external slope relocation (*Figure 3—figure supplement 1*). Given that teeth remodeling is a lengthy process that starts very early on in oceanic larvae and therefore before reef colonization, it is not surprising that the treatments performed on crest captured larvae do not affect (or are too late to affect) teeth development. Also, contrarily to individuals relocated on the outer slope, we did not observe any effect of T3 + IOP nor NH3 treatments on skin pigmentation at day 2 and day 5 (*Figure 3—figure supplement 2*). This suggests that pigmentation during the larva to juvenile transition is strongly coupled to the environment but is not directly under TH control that may not sustain such rapid pigmentation changes. Taken together: (i) intestine lengthening and remodeling are mediated by TH signaling; and (ii) disturbing the TH pathway disrupts the normal development of fish after larval recruitment, showing that TH effectively controls some major aspects of *A. triostegus* metamorphosis.

Finally, we tested how TH signaling influences fish behavior. To achieve this, we used grazing activity (number of bites on algal turf), as *A. triostegus* larvae start to graze a couple of days after colonizing the reef, to investigate how TH signaling and metamorphosis completion influence the behavior of recruited fish. T3 + IOP treatment increases the grazing activity by around 50% in day 5

juveniles when compared to the control individuals (p-value=0.004, *Figure 3G*, blue lanes). On the contrary, NH3 treatment significantly decreases the number of bites at day 2 by 33% (p-value<0.001, *Figure 3G*, green lane). The grazing activity of fish relocated on the external slope (*Figure 3A*) is also diminished by more than 4-fold at day 2 (p-value<0.001, *Figure 3G*, orange lanes). These data indicate that TH signaling promotes the biting activity of *A. triostegus* and subsequent grazing behavior adopted at larval recruitment. This shows that TH controls the transformations of the feeding process not only at the morphological and physiological levels but also at the behavioral level.

## Thyroid hormone-mediated metamorphosis is conserved in coral reef fishes

To widen our understanding of coral reef fish metamorphosis, we investigated the TH signaling during larval recruitment in four other coral fish species (*Rhinecanthus aculeatus*, *Chromis viridis*, *Chaetodon lunula* and *Ostorhinchus angustatus*) from distant families (*Near et al., 2012*). In these species, both T4 and T3 levels drop between day 1 and day 8, up to 3-fold for T4 (in *O. angustatus*, p-value=0.029) and up to 4-fold for T3 (in *R. aculeatus*, p-value=0.05) (*Figure 4*). This drop of TH levels after reef entry strongly suggests that TH are also key determinants for the metamorphosis of these species. T4 levels in *R. aculeatus* and T3 levels in *C. viridis* also rise between near ocean and day 1 individuals (*Figure 4*), further confirming the model of a TH-mediated metamorphosis during coral reef fish larval recruitment. However, T4 levels are 3-fold higher in near ocean individuals than in day 1 individuals in *C. viridis* (*Figure 4*). This suggests some potential species-specific variation in coral reef fish TH profiles at recruitment that would be extremely interesting to decipher in future studies on additional species and with regards to other aspects of coral reef fish ecology (*e.g.*, diets, size and pigmentation status at recruitment).

## Reef pollutant disrupts the transformation of *Acanthurus triostegus*

Chlorpyrifos (CPF) is an agricultural insecticide that is widely used on tropical coastal crops, therefore one of the most common waterborne chemical pollutants encountered in coral reef surrounding waters (*Cavanagh et al., 1999*; *Leong et al., 2007*; *Roche et al., 2011*; *Botté et al., 2012*). Given the endocrine determinants of coral reef fish metamorphosis (*Figures 2–3*) and the CPF endocrine disruption characteristics (*Juberg et al., 2013*; *Slotkin et al., 2013*), we assessed the impact of CPF on *A. triostegus* metamorphosis. To achieve this, we used exposure doses previously reported in coral reef fish (1, 5 or 30 $\mu$g.l$^{-1}$ of seawater) (*Botté et al., 2012*). CPF treatment at 30 $\mu$g.l$^{-1}$ decreases the T3 levels by 50% in day 2 juveniles (p-value=0.021, *Figure 5A*). No significant effect was observed on T4 levels regardless of CPF concentration and duration (p-value=0.205 for day 2 and p-value=0.496 for day 5, *Figure 5—figure supplement 1A*). We did not observe any effect of CPF treatments on fish pigmentation at day 2 and day 5 (*Figure 5—figure supplement 1B*). However, CPF exposure at 30 $\mu$g.l$^{-1}$ prevents guts' lengthening at day 2 (p-value=0.024, *Figure 5B* left panel), and both exposure at 5 and 30 $\mu$g.l$^{-1}$ prevent intestines' lengthening at day 5 (p-values=0.047 and 0.029 respectively, *Figure 5B* right panel). Strikingly, CPF exposure at 30 $\mu$g.l$^{-1}$ also decreases the grazing activity of day 2 juveniles with 2.63 ± 0.09 bites per fish per min contrasting with 3.62 ± 0.28 bites per fish per min in the solvent control condition (p-value=0.014, *Figure 5C*). In day 5 juveniles, exposure at 5 and 30 $\mu$g.l$^{-1}$ decreases the grazing activity with respectively 2.68 ± 0.18 and 2.63 ± 0.05 bites per fish per min compared to 4.43 ± 0.33 bites per fish per min for the solvent control fish (p-values=0.001, *Figure 5C*). Such a behavior in fish treated with CPF results in up to a 14-fold reduction in the biomass of turf grazed after 5 days (p-value=0.05, *Figure 5D*).

## Discussion

In this study, we investigated the morphological, physiological and behavioral changes occurring during *A. triostegus* larval recruitment. These results highlight profound anatomical and physiological transformations of the skin and the whole digestive tract (*Figure 1B–W*). In particular, the differences in 16S gut content are likely the consequence of a diet shift (carnivorous to herbivorous) and an environmental shift occurring during larval recruitment (*Frédérich et al., 2012*; *Baldo et al., 2015*; *Sullam et al., 2012*). We hypothesize that intestine remodeling (i.e. the loss of the villi and internal structures) accelerates the shift of the gut microbiota by allowing the establishment of a new bacterial community fitting the diet change. The presence of *Lachnospiraceae* in the adult but not in

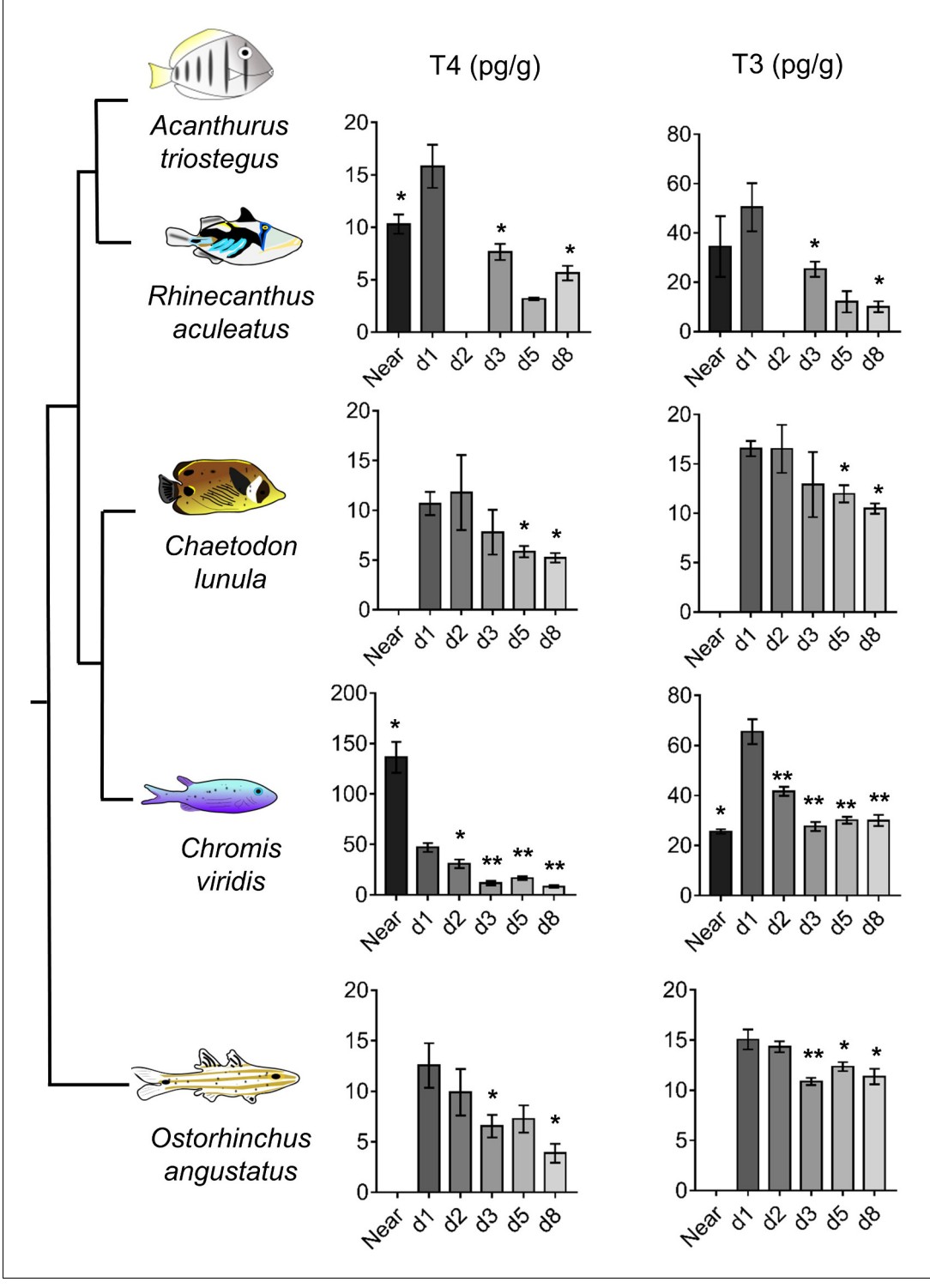

**Figure 4.** Thyroid hormone levels in distant coral reef fish at larval recruitment. Error bars represent standard errors. *p-value<0.05; **p-value<0.01; ***p-value<0.001. TH levels of: *Rhinecanthus aculeatus* (upper), *Chromis viridis* (middle up), *Chaetodon lunula* (middle low) and *Ostorhinchus angustatus* (lower). n = 3 to 5 biological replicates for each condition. First column is T4, second column is T3. Y-axis: T4 or T3 level in pg.g$^{-1}$. X-axis: sampling point (developmental stage). The phylogenic relationship between the species is plotted on the far left of the figure. Statistics: for each species, comparisons of each mean to the day 1 (d1) mean (Wilcoxon-Mann-Whitney U-test). See *Figure 4—source data 1* for detailed data and statistics.

DOI: https://doi.org/10.7554/eLife.27595.026

*Figure 4 continued on next page*

*Figure 4 continued*

The following source data is available for figure 4:

**Source data 1.** T4 and T3 levels of various coral reef fishes at different development stages.
DOI: https://doi.org/10.7554/eLife.27595.027

the larvae is consistent with the hypothesis that this family of bacteria is acquired through the environment in surgeonfishes (*Miyake et al., 2016*). Altogether, these observations reinforce the notion that the fish bacterial community depends on their diet and environment. These shifts of 16S and 18S gut contents, occurring between crest individuals and day 8 juveniles, strikingly coordinate with both the changes of the intestine and teeth morphologies observed during larval recruitment (*Figure 1M–V*). All these changes from teeth to microbiota are consistent with *A. triostegus* change of diet after entering the reefs, conforming to previous observations (*Randall, 1961*; *McCormick, 1999*; *Frédérich et al., 2012*). This complete remodeling is associated with a severe weight loss (*Figure 1B*), suggesting that larval recruitment corresponds to a real metabolic challenge.

The molecular investigation of TH signaling in the recruiting larvae indicates a surge of TH levels and *TR* expressions in near ocean larvae and crest individuals, and a drop in juveniles, after reef entry (*Figure 2A–C*). This indicates that migration towards the reefs marks the onset of metamorphosis and starts very early on (*Figure 6A*). Consequently, the transformation process is already well advanced when larvae reach the reef, suggesting that this event is triggered in the open ocean by, yet unknown, environmental cues (*Figure 6A*). After reef entry, T3 treatment increases gut length while external slope relocation and NH3 treatment delay the metamorphosis, although they could not completely interrupt it (*Figure 3A–B*). These results are consistent with the model of metamorphosis in amphibians (*Brown and Cai, 2007*) and other fish species (*McMenamin and Parichy, 2013*). It is interesting to relate the morphological and physiological changes we have observed in *A. triostegus* with the TH-controlled post-embryonic process known in zebrafish (the major teleost fish developmental model, one of the few teleost fishes in which we have a functional knowledge of the metamorphosis process). Indeed, in both species, as well as in flatfish, another classical teleost model of metamorphosis, major pigmentation and craniofacial changes occur at metamorphosis (*McMenamin and Parichy, 2013*; *McMenamin et al., 2014*). It is not clear if the appearance of the dark melanophore stripes in zebrafish and *A. triostegus* is controlled by similar mechanisms. In *A. triostegus*, this event is extremely rapid (as fast as 2 hr after reef entry, MB pers. obs.), and strongly environmentally coupled, whereas in zebrafish juveniles the pigment pattern changes very gradually and is presumably not under selection for rapid change during metamorphosis (*McMenamin et al., 2014*). However, melanophores can differentiate rapidly (in less than 12 hr) when appropriately stimulated in juvenile zebrafish. For example, acute stimulation with TH may initiate rapid melanophore differentiation (*McMenamin et al., 2014*). Therefore, if the melanophore precursors are already in place and poised to differentiate in *A. triostegus*, it is possible that they could become melanized within a few hours upon direct stimulation with TH. Our NH3 treatments suggest that this may not be the case (*Figure 3—figure supplement 2*), but this remains an open question that requires further investigation. Altogether, our findings indicate that the onset and coordination of metamorphosis are two distinct events, both coordinated by TH and under environmental control. They further demonstrate that *A. triostegus* larval recruitment is a TH-controlled metamorphosis, as in amphibians and flatfishes (*Brown and Cai, 2007*; *Laudet, 2011*; *McMenamin and Parichy, 2013*).

We demonstrated that the decrease of both T4 and T3 after reef entry can be generalized to phylogenetically distant coral fish species (*Figure 4*). We propose that this decrease is valid for most if not all coral reef fishes, even if species-specific variations on this general theme can be observed, and that the larval recruitment of coral reef fish corresponds to a TH-controlled metamorphosis (*Figure 6A*). Interestingly, variations in the magnitude of TH changes among the five coral reef fish species (*Figure 4*) are reminiscent of the situation in stickleback, where differences in TH signaling between two ecotypes support life history trait variation (*Kitano et al., 2010*; *Kitano and Lema, 2013*). We anticipate that this post-embryonic developmental step is shaped by evolution and varies according to species-specific life history traits such as pelagic larval duration, size at reef colonization and pigmentation status (i.e., metamorphic state of advancement) at recruitment. Given the

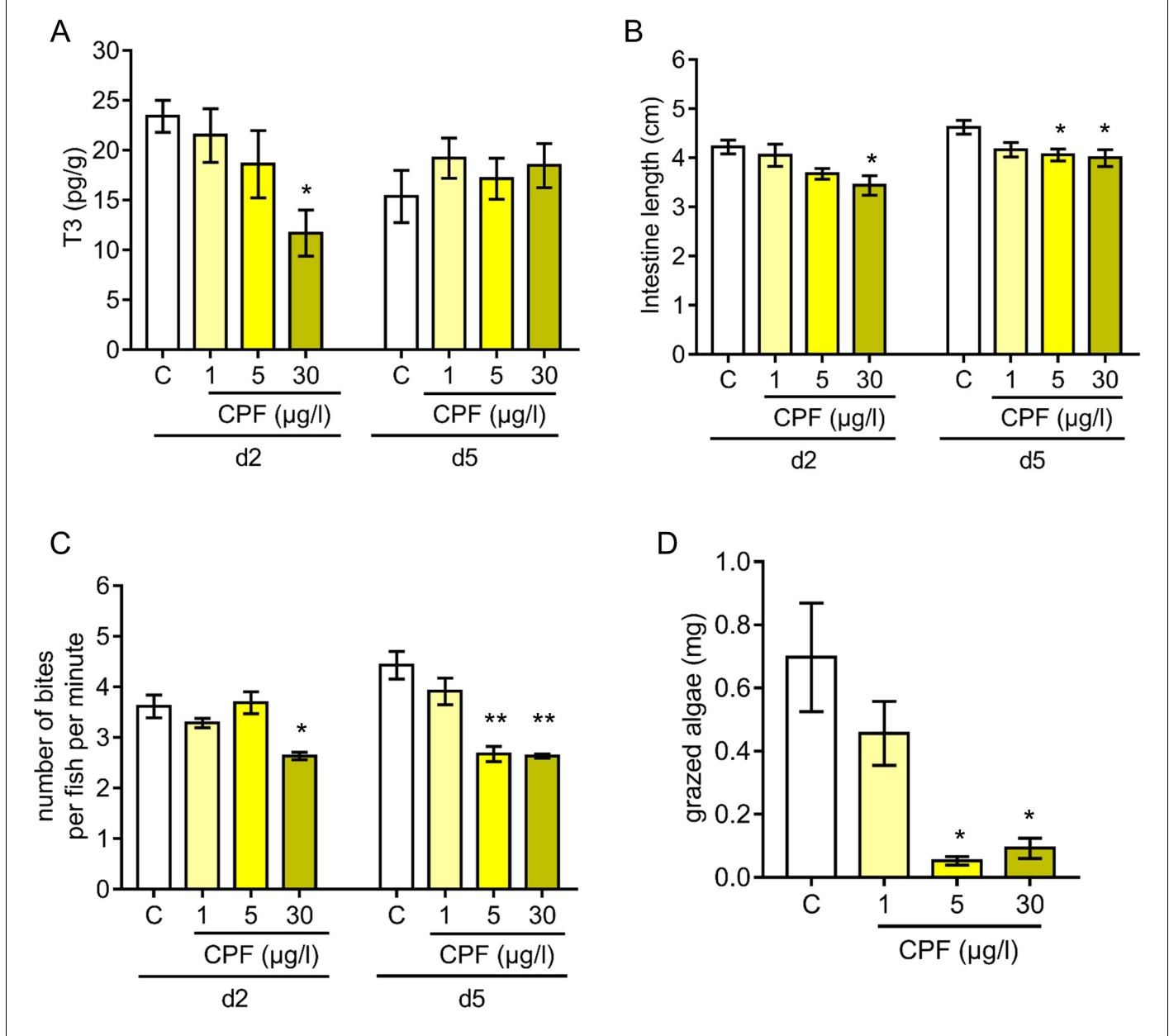

**Figure 5.** Influence of chlorpyrifos exposure on *Acanthurus triostegus* thyroid hormone signaling, intestine length, grazing activity, and algal removal efficiency. Error bars represent standard errors. *p-value<0.05; **p-value<0.01; ***p-value<0.001. (A) T3 level in pg.g$^{-1}$ of day 2 (d2) and day 5 (d5) fish exposed to different conditions: solvent control (C, acetone, white bars), or exposed to CPF at 1 µg.l$^{-1}$ (1, light yellow bars), 5 µg.l$^{-1}$ (5, yellow bars) or 30 µg.l$^{-1}$ (30, dark yellow bars). Exposure started right after crest capture. n = 10 fish in each condition. Statistics: ANOVA followed by Tukey posthoc tests, only significant differences with the solvent-control condition are indicated. See *Figure 5—source data 1* for detailed data and statistics. (B) Intestine length in cm of fish at day 2 (d2) or day 5 (d5) under different conditions: solvent control (C, acetone, white bars), or exposed to CPF at 1 µg. l$^{-1}$ (1, light yellow bars), 5 µg.l$^{-1}$ (5, yellow bars) or 30 µg.l$^{-1}$ (30, dark yellow bars). Exposure started right after crest capture. n = 5 to 8 fish for each experimental condition. Statistics: same as in (A). See *Figure 5—source data 2* for detailed data and statistics (C) Mean number of bites (on turf algae) per fish in a batch of 18 fish on reef turf at day 2 (d2) or day 5 (d5), under different conditions: solvent control (C, acetone, white bars), or exposed to CPF at 1 µg.l$^{-1}$ (1, light yellow bars), 5 µg.l$^{-1}$ (5, yellow bars) or 30 µg.l$^{-1}$ (30, dark yellow bars). Exposure started right after crest capture. n = 3 independent bite counts for each condition. Statistics: same as in (A). See *Figure 5—source data 3* for detailed data and statistics. (D) Mean turf biomass removed per fish in a batch of 10 fish, in 5 days following reef colonization, under different conditions: solvent control (C, acetone, white bars), or exposed to CPF at 1 µg.l$^{-1}$ (1, light yellow bars), 5 µg.l$^{-1}$ (5, yellow bars) or 30 µg.l$^{-1}$ (30, dark yellow bars). Exposure started right after crest capture. n = 3 independent measurements of grazed turf for each condition. Statistics: comparisons of each mean to the solvent-control (C) mean (Wilcoxon-Mann-Whitney U-test). See *Figure 5—source data 4* for detailed data and statistics.

DOI: https://doi.org/10.7554/eLife.27595.028

*Figure 5 continued on next page*

*Figure 5 continued*

The following source data and figure supplement are available for figure 5:

**Source data 1.** T3 levels in *Acanthurus triostegus* under different chlorpyrifos exposure conditions.
DOI: https://doi.org/10.7554/eLife.27595.030

**Source data 2.** Intestine length of *Acanthurus triostegus* under different chlorpyrifos exposure conditions.
DOI: https://doi.org/10.7554/eLife.27595.031

**Source data 3.** Grazing activity of *Acanthurus triostegus* under different chlorpyrifos exposure conditions.
DOI: https://doi.org/10.7554/eLife.27595.032

**Source data 4.** Quantity of grazed algae by *Acanthurus triostegus* under different chlorpyrifos exposure conditions.
DOI: https://doi.org/10.7554/eLife.27595.033

**Source data 5.** T4 levels in *Acanthurus triostegus* under different chlorpyrifos exposure conditions.
DOI: https://doi.org/10.7554/eLife.27595.034

**Figure supplement 1.** Pigmentation patterns and chlorpyrifos exposure in *A. triostegus* at metamorphosis.
DOI: https://doi.org/10.7554/eLife.27595.029

dramatic diversity of life history strategies they propose, coral reef fish could be excellent models to decipher how these changes are elicited by changes in hormonal systems.

Among the endocrine disruptors already evidenced and encountered in coral reefs, CPF is known to be a thyroid disruptor with adverse effects on fish behavior (*Sandahl et al., 2005*; *Kavitha and Rao, 2008*; *Roche et al., 2011*; *Yang et al., 2011*; *Botté et al., 2012*; *Deb and Das, 2013*; *Juberg et al., 2013*; *Slotkin et al., 2013*). We demonstrated that CPF exposure reduces *A. triostegus* physiological level of T3 at larval recruitment (*Figure 5A*). Consistently with our observations on NH3 treated fish, exposure to CPF impairs metamorphosis, as evidenced with the repression of guts' lengthening (*Figure 5B*), and the decrease in the juveniles' grazing activity (*Figure 5C*). This latter point led to a reduced algal removal efficiency (*Figure 5D*). As *A. triostegus* is one of the major herbivorous fish in coral reefs, this can severely impair the ability of coral reefs to recover after an algal invasion (e.g., after a bleaching event) and therefore contributes to the worldwide degradation of reef health (*Russ, 1984*; *Almany et al., 2007*; *Ledlie et al., 2007*; *Hughes et al., 2007*; *Hughes et al., 2014*) (*Figure 6B*).

## Conclusion

Our results highlight how coral reef fish larval recruitment, the transition between the open ocean and the reef, is a TH-regulated metamorphosis at the crossroads of ecological, developmental, physiological and behavioral transformations (*Holzer and Laudet, 2015*). These findings also demonstrate how coral reef fish TH signaling and larval recruitment processes can be altered by reef pollution. Since metamorphosis and larval recruitment are essential for the maintenance of fish populations and subsequent coral reef resilience, this study provides a general framework to better understand, at the molecular level, how global changes, water pollution and human activities can threaten reef ecosystems.

## Materials and methods

### Fish capture, husbandry and conservation

*Acanthurus triostegus*, *Ostorinchus angustatus*, *Chaetodon lunula*, *Chromis viridis* and *Rhinecanthus aculeatus* larvae were captured at Moorea Island, French Polynesia. Larvae were captured during the night while colonizing the reef crest, using a crest net and hand nets (*Dufour and Galzin, 1993*; *Besson et al., 2017*). Larvae were then kept in cages located in situ in the lagoon and provided with continuous food from the water column supplemented with coral rubble and algal turf. For the clarity of this study, the age in day of the juvenile corresponds to the number of day spend in the reef in cages since capture, not the absolute age. Near ocean *A. triostegus* larvae were sampled at 1–2 km away from the reef crest using drifting light trap and hand nets from 0.5 m to 5 m depth. Far ocean *A. triostegus* larvae were also captured offshore by trawl haul more than 10 km away from the reef between 1 and 60 m deep. Larvae were euthanized in MS222 at 0.4 mg.ml$^{-1}$ in filtered seawater. For TH dosage, larvae were frozen dry and conserved at $-20°C$ prior to extraction. For RNA

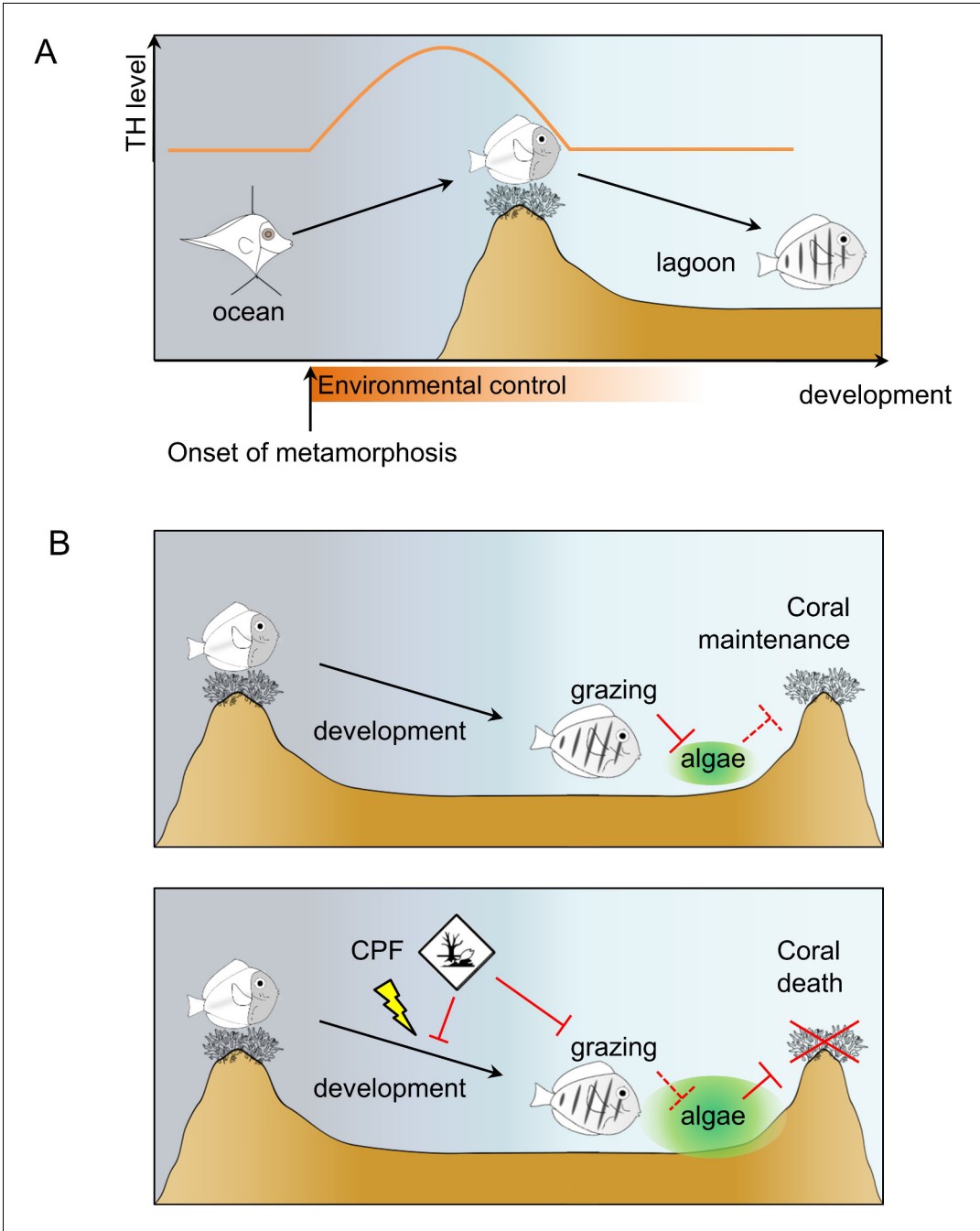

**Figure 6.** Model of *Acanthurus triostegus* metamorphosis and disruption by chlorpyrifos. (**A**) Schematic representation of coral reef fish larval recruitment. TH level is plotted on the Y axis and development on the X axis. The onset of metamorphosis is in the open ocean and environmental influence upon development is plotted on the X-axis. (**B**) Consequences of undisturbed (upper panel) and CPF disturbed (lower panel) metamorphosis on grazing activity, and subsequent algal spreading and potential coral survival. Solid red bar indicates inhibition; dotted red line indicates lack of inhibition.

DOI: https://doi.org/10.7554/eLife.27595.035

extraction, larvae were lacerated and kept in RNAlater (*Sigma*) 1 hr on ice and stored at −20°C. For histology, larvae were rinsed in PBS 1X, dissected and the intestines were kept in Bouin's fixative at room temperature. For µCT scan and metagenomics, Larvae were kept in 70% ethanol at room temperature.

## Fish treatments

*Acanthurus triostegus* captured at the crest were injected with 20 µl of drugs in the ventral cavity. Drugs tested were as follows: (i) solvent control (DMSO diluted 10.000 times in Phosphate Buffer Saline 1X, as all drugs were made soluble in DMSO and diluted 10.000 in PBS 1X); (ii) T3 + iopanoic acid (IOP) both at $10^{-6}$ M,; (iii) NH3 at $10^{-6}$ M; and (iv) T3 + IOP + NH3 all at $10^{-6}$ M. IOP was used as an inhibitor of deiodinase enzymes, as evidenced in mammals and amphibians (*Galton and Hiebert, 1987*; *Galton and Hiebert, 1988*; *Galton, 1989*; *Simonides et al., 2008*; *Medina et al., 2011*; *Renko et al., 2012*), and as routinely used in fish to prevent the immediate degradation of injected T3 (*Little et al., 2013*; *Lorgen et al., 2015*). NH3 is a known TR antagonist in vertebrates (*Lim et al., 2002*), a notion we confirmed in *A. triostegus* (*Figure 2—figure supplement 1D*). NH3 prevents the binding of TH on TR and impairs the binding of transcriptional coactivators to TR, which therefore remain in inactive conformation (*Figueira et al., 2011*; *Lim et al., 2002*). Fish were then kept in the lagoon in in situ cages and injected daily following the same protocol to maintain pharmacological treatment. Fish were sampled on the second and fifth day after the beginning of this protocol. DMSO non toxicity was tested by comparing the pigmentation, the teeth development, the intestine length and internal structures between solvent-control individuals (treated with DMSO diluted 10.000 times in PBS 1X), and control individuals (simply raised in situ in lagoon cages as mentioned in the previous paragraph) (*Figure 3—figure supplement 1*, *Figure 3—figure supplement 2*).

## Chlorpyrifos (CPF) exposure

After crest capture, *A. triostegus* larvae were transferred in aquaria (30 L x 20 W x 20 cm H) in groups of n = 10 fish. Each aquarium was filled with 9 liters of UV-sterilized and filtered - 10 µm filter, and was equipped with an air stone. Fish were then immediately exposed to: nothing (control), acetone at a final concentration of 1:1.000.000 (solvent control, as CPF was made soluble in acetone) or CPF at a final concentration of 1, 5, and 30 µg.l$^{-1}$, as previously done in the literature in another coral reef fish (*Botté et al., 2012*). All water was replaced every day, ensuring the maintenance of water quality as well as a continuous concentration of the pesticide or solvent. Fish were sampled the 2$^{nd}$ and 5$^{th}$ day after the beginning of this protocol. Acetone non toxicity was tested by comparing the TH levels, the pigmentation, and the intestine length between solvent-control and control individuals (*Figure 5A–B*, *Figure 5—figure supplement 1A–B*).

## Cloning of *Acanthurus triostegus* genes

Muscle from adult fish and whole larvae were used. Samples were cut using sterile scalpel blades and crushed in a Precelyss with Qiagen extraction buffer. RNA extractions were performed using the Macherey-Nagel RNA extraction kit following the manufacturer's instructions. Total pooled RNAs were retro-transcribed with the Invitrogen Super Script III enzyme following the manufacturer's instructions. For *TR* cloning, actinopterygian universal primers (*Figure 2—figure supplement 2*) were designed for *TRα-A*, *TRα-B* and *TRβ* to retrieve the full-length sequences of the genes. For *polD2* and *rpl7*, actinopterygian universal primers were designed to retrieve partial sequences including at least one exon-exon junction. PCR amplicon were cloned in the Invitrogen PCR II plasmid following the manufacturer instruction for sequencing. *TRα-A*, *TRα-B* and *TRβ* were subcloned in pSG5 between EcoRI sites for *in cellulo* expression.

## qPCR assay

Far ocean to day 8 juveniles were assayed. For each larvae 1 µg of RNA were used for retro-transcription using the *Invitrogen Super Sript III* following the manufacturer's instructions, including a DNAse I treatment. qPCR primers were designed to anneal on different exons, R*pl7* and P*old2* were used as normalization genes (*Figure 2—figure supplement 2*). qPCRs were performed in 96 well plate with the *BioRad IQ Syber Green Super Mix* in 10 µl of final reaction per well following the manufacturer's ratios. qPCRs were assayed in a BioRad thermocyclers and analysed on *BioRad CFX Manager software*. Assays were performed on duplicates in at least two independent RNA extractions and retro-transcriptions. *Klf9*, which is known to respond to TH signaling, was used as a TH signaling reporter gene.

## Phylogenetic analysis

The amino acid sequences of *A. triostegus* cloned TRs as well as available TR sequences (*Figure 2— figure supplement 3*) were aligned using *MUSCLE* software. Trees were generated using the Maximum Likelihood method with the *Seaview four* software under the *JTT model* with estimated gamma shape and eight rate categories (RRID:SCR_015059) (*Gouy et al., 2010*). Bootstrap analysis of 1000 replicates was carried out to support the tree.

## Functional characterization of the receptors

Human embryonic kidney 293 cells (ATCC:CRL_11268, RRID:CVCL_0045) (*Iwema et al., 2007*; *Gutierrez-Mazariegos et al., 2014*; *Sadier et al., 2015*) were grown in Dulbecco's modified Eagle's medium supplemented with 10% of coal stripped foetal bovine serum and penicillin/streptomycin at 100 µg.ml$^{-1}$. Cell were maintained at 37°C, 5% $CO_2$ and tests for mycoplasma contamination were negative. The transient transfection assays were carried out in 96-well plate with 30 000 cells per plate using *Exgen500* according to the manufacturer's instructions. For each well, cells were transfected with 50 ng of total DNA: 12,5 ng of full-length receptor encoding plasmid, 12,5 ng of reporter plasmid with four DR4 repeat in the luciferase promoter, 12,5 ng of β-galactosidase encoding plasmid and 12,5 ng of *pSG5* empty plasmid. Drugs were incubated for 48 hr and cells were harvested using a passive lysis buffer and frozen at −20°C. On half of the lysate, luciferase activities were assayed with the luciferase reagent buffer from *Promega* on a *Veritas Turner Biosystem luminometer*. On the other half of the lysate, the β-galactosidase activity was measured using ONPG substrate and absorbance at 420 nm for normalization. Each assay was performed at least three times independently on well triplicates. Drugs from *Sigma-Aldrich* were diluted in DMSO à 10$^{-2}$ M then in sterile PBS1X prior treatment.

## Thyroid hormone quantification

TH were extracted from dry-frozen fish following an extraction protocol adapted from previous publications relating TH level variations in teleost fish (*Tagawa and Hirano, 1989*; *Einarsdóttir et al., 2006*; *Kawakami et al., 2008*). Far ocean larvae to day 8 juveniles were assayed. At least three individuals per sampling point were used. Each fish was first crushed with a *Precelyss* in 500 µl methanol, centrifuged at 4°C and supernatant reserved, three times. Pooled supernatant were dried at 70°C. Hormones were re-extracted with 400 µl methanol, 100 µl chloroform and 100 µl barbital buffer twice from the first dried extract. Pooled supernatant was dried out and extract was reconstituted in 2 ml of PBS1X for quantification. The quantification was performed following the *Roche ELICA* kit on a Cobas analyser by a medical laboratory according to the manufacturer's standardized method.

## Intestine histology

The intestines in Bouin's fixative were embedded in paraffin. Sections of 5 µm were performed in the proximal, medial and distal part using a microtome (*Leica*). The histological sections were stained with Hematoxylin and Eosin. Photography were taken on a Leica microscope and the mosaic reconstructed using Image J software.

## Gut metagenomics

For 16S mass sequencing, juveniles of 2, 5, 8 days after reef entry and adult *A. triostegus* gut were dissected in triplicates for a total of twelve samples. Total DNA was extracted with a *Macherey-Nagel* DNA extraction kit following the manufacturer's recommendations. 16S library were constructed for each individual using the *Ion 16S Metagenomic* kit from *Life technologies* following the manufacturer's recommendations. Sequencing of bacterial 16S was performed using a PGM Ion Torrent. Three controls for contaminations were performed. The environment control consisted of an open tube during the dissection. The extraction control monitored the DNA extraction process. The blank control monitored the library construction. The sequencing results were then analyzed using the *Life technologies* 16S pipeline. 18S mass sequencing was performed on crest larvae, 8 days juveniles and adults in triplicates. Intestine contents were extracted in the same condition as 16S. 18S library were constructed with the pre-amplified V7 region of the eukaryotic 18S (*Figure 2—figure supplement 2*). A blocking primer was designed to prevent the amplification of host sequences (*Figure 2—figure supplement 2*). The blocking primer was designed to partially overlap the 18S reverse

primer and was modified with a C3 spacer. Sequencing was performed using a *PGM Ion Torrent*. The sequences were assembled and blasted against the PR2 database, completed with 18S sequenced from multi-cellular organisms for taxonomic affiliation.

### μCT scan analysis

Fish samples were conserved in 70% ethanol, dehydrated in successive baths of 95% ethanol, twice 100% ethanol and in vacuum arena for at least 4 hr. X-ray microtomographies were performed on a *Phoenix Nanotom (General Electric)* at 70 kV of tension, 100 mA of intensity with a tungsten filament. 3000 images per sample were taken at 500 ms of exposure per image and at a resolution ranging from 2.5 to 2.8 μm. 3D volumes were reconstructed and analysed with VGI studiomax software.

### Grazing activity

For each condition (e.g. hormonal treatments, pesticide conditionings), 18 fish were placed in a 5 liter tank with coral rubbles covered with turf algae. After an acclimation period of 1 hr, we recorded the total number of bites made on turf algae during 10 min. This experiment was replicated with a new batch of fish 3 to 6 times depending on the condition.

### Grazed turf biomass

For each pesticide exposition conditioning, we weighed (underwater) 3 pieces of coral rubble prior the introduction of a group of 20 crest captured fish (weight A), and after 5 days of grazing by this group of fish (weight B). For each piece of coral rubble, the weight of grazed algae was estimated through the difference weight B minus weight A.

### Statistical information

All statistical analyses were conducted using the *R-Cran* project free software (http://www.rproject.org/, *R-3.3.1*). Mean comparisons were performed using Wilcoxon-Mann-Whitney U-test when comparing two means, and using univariate analysis of variance (ANOVA) followed by Tukey post-hoc test (should a significant difference be detected) for multiple comparisons. Prior ANOVA, normality of values (or residuals) and variance homogeneity were assessed using Shapiro and Bartlett tests. In qPCR analyses, comparisons of gene expression were performed through Student's t cumulative distribution functions, automatically computed by *qPCR software CFX Biorad Manager*, (*CFX Manager SH, 2017*). All statistical information and data are available in each Figure source data (see Figure captions).

## Acknowledgement

We thank M Bronner, S McMenamin, D Hazlerigg and A Schreiber for their comments that greatly improved this manuscript. We would also like to thank H Jacob, C Gache, A Gilson, I Moniz, C Berthe, A Savura, D Van Osten and P Schep for their help to catch fish larvae. We thank S Planes, J Burden, T Lorin, M Manceau, P Salis and N Roux for critical reviewing and for their constructive discussion concerning coral fish ecology and development. We are grateful to V Domien for her fruitful advice on the design of the figures. We thank the association *Te mana o te moana* for the accommodation of our cages.

## Additional information

### Funding

No external funding was received for this work

### Author contributions

Guillaume Holzer, Conceptualization, Data curation, Formal analysis, Validation, Investigation, Visualization, Methodology, Writing—original draft, Writing—review and editing, Wrote the paper, Designed the experiments, Performed gene cloning, qPCR, Transactivation assay, μCT scan, Hormone dosage, Assisted in metagenomics and analyzing the data, Assisted with preparing the

manuscript; Marc Besson, Conceptualization, Data curation, Formal analysis, Supervision, Validation, Investigation, Visualization, Methodology, Writing—original draft, Writing—review and editing, Wrote the paper, Designed the experiments, Performed hormone dosage, Fish capture, Fish treatment, Grazing experiments, Histology and analyzing the data, Assisted with preparing the manuscript; Anne Lambert, Data curation, Investigation, Visualization, Methodology, Assisted with histology and data analysis, Assisted with preparing the manuscript; Loïc François, Conceptualization, Data curation, Investigation, Visualization, Writing—review and editing, Assisted with fish capture, fish treatment and analysis of the data, Assisted with preparing the manuscript; Paul Barth, Data curation, Formal analysis, Assisted in gene cloning, Hormone dosage, Analysis of the data, Assisted with preparing the manuscript; Benjamin Gillet, Data curation, Formal analysis, Validation, Investigation, Visualization, Methodology, Writing—review and editing, Performed the metagenomics, Design of the experiment and assisted in the analysis of the data, Assisted with preparing the manuscript; Sandrine Hughes, Data curation, Formal analysis, Validation, Investigation, Visualization, Methodology, Writing—review and editing, Performed the metagenomics analysis, Assisted in the experiment and the design, Assisted with preparing the manuscript; Gwenaël Piganeau, Data curation, Formal analysis, Visualization, Writing—review and editing, Assisted in the analysis of metagenomics data, Assisted with preparing the manuscript; Francois Leulier, Conceptualization, Data curation, Formal analysis, Validation, Investigation, Visualization, Methodology, Writing—review and editing, Assisted in the analysis of metagenimic data and design of the experiment, Assisted with preparing the manuscript; Laurent Viriot, Performed and analyzed X-ray microtomography, Read and commented on the manuscript; David Lecchini, Conceptualization, Resources, Data curation, Formal analysis, Supervision, Validation, Investigation, Visualization, Methodology, Writing—original draft, Project administration, Writing—review and editing, Wrote the paper, Designed the experiments, Performed fish capture, Analyzed the data, Assisted with preparing the manuscript; Vincent Laudet, Conceptualization, Resources, Data curation, Formal analysis, Supervision, Validation, Investigation, Visualization, Methodology, Writing—original draft, Project administration, Writing—review and editing, Wrote the paper, Designed the experiments

## Author ORCIDs

Marc Besson https://orcid.org/0000-0003-3381-322X

## Ethics

Animal experimentation: This study was performed in strict accordance with the guidelines of the French Polynesia committee for publication and animal ethics. All the animals were handled in accordance with the guidelines CRIOBE-IRCP animal ethics committee, and every effort was made to minimize suffering. The experiments and protocols were approved by the CRIOBE-IRCP animal ethics committee (Decision letter 02-14-2015)

## Decision letter and Author response

Decision letter https://doi.org/10.7554/eLife.27595.037
Author response https://doi.org/10.7554/eLife.27595.038

# Additional files

## Supplementary files

• Transparent reporting form
DOI: https://doi.org/10.7554/eLife.27595.036

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
