## [Decision Letter]

Thank you for submitting your article "Fish larval recruitment is a metamorphosis sensitive to reef pollutants" for consideration by *eLife*. Your article has been reviewed by three peer reviewers, and the evaluation has been overseen by Marianne Bronner as the Senior Editor and Reviewing Editor. The following individuals involved in review of your submission have agreed to reveal their identity: Sarah K McMenamin (Reviewer #1); David Hazlerigg (Reviewer #2); Alexander Schreiber (Reviewer #3).

The reviewers have discussed the reviews with one another and the Reviewing Editor has drafted this decision to help you prepare a revised submission.

Summary:

This paper considers the role of thyroid hormone in larval development in reef dwelling fish, with an emphasis on endocrine disruption and possible impacts on ecosystem health. Although the implication of TH in metamorphic change in reef dwelling fish is not surprising, the predicted impacts in terms of reef grazing are quite impressive. Thus, this is an important and timely study, one that successfully integrates endocrine, developmental, ecological, and toxicological influences on a key life history transition in a reef fish. Much of the text is cumbersome to read due to English not being the primary language of the authors. Specific required changes are listed below.

Essential revisions:

1) On the drug / endocrine disruptor treatments, the authors are vague about mode of action. E.g. iopanoic acid (IPA) is described as "an inhibitor of T4 degrading enzymes": actually IPA inhibits TH deiodinases, which act on both T4 and T3, and at least in mammals it is selective for outer ring deiodination by dio2. In fish the picture is probably less clear, due to uncertainties about pharmacological selectivity. Some more effort at clarity in this direction – not only for IPA but also for the other compounds employed – would be welcome.

2) By far the most interesting aspect of the study is the determination that larval recruitment in this species is indeed a TH-mediated process that induces diverse morphological and behavioral changes. The effects of chlorpyrifos on metamorphosis are interesting, but it is certainly a stretch to imply that chlorpyrifos is representative of all pollutants (there are many pollutants that likely don't affect TH, but instead affect other pathways, such as estrogen signaling). Therefore, the title should be revised as follows: "Larval fish recruitment to reefs is a thyroid hormone-mediated metamorphosis sensitive to the pesticide chlorpyrifos".

3) The various morphological and histological analyses performed on natural *A. Triostegus* metamorphosis are excellent, and the changes in the gut are particularly fascinating! Considering this, it is surprisingly that the authors did not report the effects of T3/NH3 treatment or of CPF on any of these morphological features. Do these compounds have any effect on cross-sectional development of the gut, pigmentation, or dentition? If so, this should be reported and shown in a figure/table.

4) Please add more discussion of the gut 16S sequencing results. What does this imply about the ecology of larval vs. adult fish, and how does it integrate with the changes in craniofacial/tooth morphology and gut structure?

5) Are there any changes in pigment pattern, dentition or gut morphology in goitrogen-treated fish relative to controls? These should be shown, or if there are no changes, the lack of changes should be discussed.

6) Are there ocean samples for the other species examined in Figure 4? It would be valuable to show the increase in TH as fish approach the crest.

7) Are there any phenotypic aspects of metamorphosis that are blocked with CPF treatment?

---

## [Author Response]

Essential revisions:1) On the drug / endocrine disruptor treatments, the authors are vague about mode of action. E.g. iopanoic acid (IPA) is described as "an inhibitor of T4 degrading enzymes": actually IPA inhibits TH deiodinases, which act on both T4 and T3, and at least in mammals it is selective for outer ring deiodination by dio2. In fish the picture is probably less clear, due to uncertainties about pharmacological selectivity. Some more effort at clarity in this direction – not only for IPA but also for the other compounds employed – would be welcome.

The reviewers correctly pointed out our mistake in presenting the iopanoic acid (that we abbreviate IOP but that is abbreviated IPA by the reviewers) as “an inhibitor of T4 degrading enzyme”. We have corrected this mistake and changed this text into: “IOP being used as an inhibitor of the deiodinase enzymes”. To introduce the deiodinases to the reader we have also added the following sentence and reference in the Introduction: “The transformation of T4 into T3, and the degradation of T4 and T3 are controlled by a family of enzymes called deiodinases (Bianco and Kim 2006).” Concerning the selectivity of IOP (or IPA) for the outer ring deiodination performed by dio2, we are not aware of such a specificity. We have found some research discussing the effective inhibition of dio3 by IOP in mammals (Simonides et al. 2008; Medina et al. 2011). Moreover, in amphibians, both the inner ring and outer ring deiodination are inhibited by IOP, indicating that all the types of deiodinases are inhibited by IOP (Galton and Hiebert 1987; Galton and Hiebert 1988; Galton 1989). Nevertheless, we agree that deiodinases are less known in fish than in mammals and amphibians, because of fewer research. In our study, we used IOP to prevent the rapid degradation of T3 in the injected fish. Such treatments are routinely performed in fish to inhibit deiodinases (Little et al. 2013; Lorgen et al. 2015). Given the existing literature about IOP and its use, we believe that adding IOP to T3 can effectively prevents T3 degradation to ensure that the treatment exert a biological activity. We have added these clarifications in the Results section: “T3 + iopanoic acid (IOP) both at 10^-6^ M, IOP being used as an inhibitor of the deiodinase enzymes that therefore prevents the degradation of injected T3 (Galton 1989; Little et al. 2013; Lorgen et al. 2015)”, and in the Materials and methods section: “IOP was used as an inhibitor of deiodinase enzymes, as evidenced in mammals and amphibians (Galton and Hiebert 1987; Galton and Hiebert 1988; Galton 1989; Simonides et al. 2008; Medina et al. 2011; Renko et al. 2012), and as routinely used in fish to prevent the immediate degradation of injected T3 (Little et al. 2013; Lorgen et al. 2015)”.

The other compound that we used is NH3, a TR antagonist that binds mammalian TR with a nanomolar affinity (Figueira et al. 2011). This compound works by preventing the binding of TH on TR (by occupying the pocket to which T3 normally binds) therefore impairing the recruitment of coactivators on TR, which thus remain in an inactive conformation (Figueira et al. 2011). To confirm the efficiency of NH3 as an antagonist on *A. triostegus TRs*, we have added, as a new panel D in Figure 2—figure supplement 1, the results of a transactivation assayexperiment. This experiment shows that, in *A. triostegus*, the TR transcriptional activity induced by T3 is competitively inhibited in presence of NH3, in a dose dependant manner. Consequently, we have also added this clarification about NH3 in the Results section: “NH3 at 10^-6^ M, NH3 being used as an antagonist of TR that prevents the binding of TH on TR (Lim et al. 2002; Renko et al. 2012) therefore disrupting *A. triostegus* TH pathway (Figure 2—figure supplement 1)” and in the Materials and methods section: “NH3 is a known TR antagonist in vertebrates (Lim et al. 2002), a notion we confirmed in *A. triostegus* (Figure 2—figure supplement 1). NH3 prevents the binding of TH on TR and impairs the binding of transcriptional coactivators to TR, which therefore remain in inactive conformation (Figueira et al. 2011; Lim et al. 2002).”.

We did not use goitrogens in this study, indeed goitrogens are inhibitor of TH synthesis and given that the TH level are already high in the crest captured larvae and decrease in the following days, we believe that goitrogen treatment would not be appropriate.

2) By far the most interesting aspect of the study is the determination that larval recruitment in this species is indeed a TH-mediated process that induces diverse morphological and behavioral changes. The effects of chlorpyrifos on metamorphosis are interesting, but it is certainly a stretch to imply that chlorpyrifos is representative of all pollutants (there are many pollutants that likely don't affect TH, but instead affect other pathways, such as estrogen signaling). Therefore, the title should be revised as follows: "Larval fish recruitment to reefs is a thyroid hormone-mediated metamorphosis sensitive to the pesticide chlorpyrifos".

We agree with the reviewers. We have changed the manuscript title accordingly to the reviewers’ recommendations: "Fish larval recruitment to reefs is a thyroid hormone-mediated metamorphosis sensitive to the pesticide chlorpyrifos".

3) The various morphological and histological analyses performed on natural A. Triostegus metamorphosis are excellent, and the changes in the gut are particularly fascinating! Considering this, it is surprisingly that the authors did not report the effects of T3/NH3 treatment or of CPF on any of these morphological features. Do these compounds have any effect on cross-sectional development of the gut, pigmentation, or dentition? If so, this should be reported and shown in a figure/table.

Following reviewers’ recommendations we have conducted additional analyses in order to answer these concerns. These analyses have been performed on additional samples that we possessed since our initial sampling campaigns, and on new samples that we captured during a new sampling campaign dedicated to these revisions. As suggested we have looked for effect of these compounds on teeth and guts development, as well as pigmentation.

First, we did not observe any difference in teeth development between larvae treated with T3, NH3, or relocated on the external slope (see Figure 3—figure supplement 1, first column). Actually, it is not surprising that neither 2 to 5 days hormonal treatments nor environmental conditioning affect this developmental process, since teeth remodeling starts very early on in oceanic larvae, even before they enter in the reef (see Figure 1).

Furthermore this remodeling is a lengthy process (see Figure 1). We have discussed this point in our manuscript by adding: “We did not observe any effect on teeth development neither with T3+IOP or NH3 treatment, nor with external slope relocation (Figure 3—figure supplement 1). Given that teeth remodeling is a lengthy process that starts very early on in oceanic larvae and therefore before reef colonization, it is not surprising that the treatments performed on crest captured larvae do not affect (or are too late to affect) teeth development”.Consequently to this observation, we did not proceed teeth morphology analyses in fish exposed to chlorpyrifos (CPF), as it seems unlikely that such exposure could lead to any teeth remodeling difference given that hormonal treatments did not.

Second, concerning pigmentation, black stripes appear rapidly in *A. triostegus* after larvae enter the reef (in 2 to 4 hours post-settlement, MB pers. obs.), and the white body pigmentation between the black vertical stripes appears at day 2 after reef entry (McCormick 1999). A clear pigmentation delay was observed in fish relocated on the external slope, as evidenced by McCormick (1999). We have presented this result more clearly: “In order to study how the environment controls the metamorphosis processes, we first relocated crest-captured larvae back to the external slope immediately after their reef entry. […] Similar to this earlier study, we observed a striking delay in the white pigmentation appearance between the vertical black stripes in day 2 relocated fish compared to their lagoon raised counterparts (Figure 3, lower panel).”

However, we did not observe any obvious difference in pigmentation between control larvae and larvae treated with T3+IOP, NH3 or exposed to CPF, both at day 2 and day 5. We have provided two additional figures (Figure 3—figure supplement 2 and Figure 5—figure supplement 1) to present these observations that we have added in the Results section as follows: “Also, contrarily to individuals relocated on the outer slope, we did not observe any effect of T3+IOP nor NH3 treatments on skin pigmentation at day 2 and day 5 (Figure 3—figure supplement 2). This suggests that pigmentation during the larva to juvenile transition is strongly coupled to the environment but is not directly under TH control that may not sustain such rapid pigmentation changes” and: “We did not observe any effect of CPF treatments on fish pigmentation at day 2 and day 5 (Figure 5—figure supplement 1).” We suspect other hormonal systems to be at play here (cortisol in particular) and we are planning to test this hypothesis in a future study.

Finally, concerning the gut histology, we have performed and analyzed cross-sections of guts in fish treated with T3+IOP or NH3, as well as in fish relocated on the external slope. We observed interesting effects of these treatments on the microvilli remodeling during metamorphosis and we have added two additional figures to present these results (see Figure 3 and Figure 3—figure supplement 1). Indeed, we observed an early disappearance of microvilli in T3+IOP treated fish at day 2 post-settlement (in comparison to the control group, Figure 3, second column, first row) and an advanced development of the second batch of microvilli (accompanied with an advanced thickening of the gut epithelium) in T3+IOP treated fish at day 5 post-settlement (Figure 3, second column, second row). At the opposite, NH3 treatment and external slope relocation both prevented guts remodeling at both day 2 and day 5, as evidenced in Figure 3 (third and fourth column). These changes are perfectly consistent with the notion that guts remodeling at metamorphosis (both the lengthening and the microvilli renewal) are controlled by TH.

We have presented and explained these results in the Results section as follows: “A similar pattern was observed concerning the microvilli remodeling within guts. […] These changes are consistent with the notion that TH control remodeling of the guts at metamorphosis in *A. triostegus.*”

Please note that for reading convenience, the former Figure 3, is now the Figure 3 as the guts cross-sections have been inserted as the new Figure 3. Also former Figure 3 is now Figure 3 as Figure 3 have been relabeled Figure 3 to present the experimental setup of Figure 3 earlier in the text (which lead us to also present the results of the external slope relocation earlier in the text, and therefore to relabel all the panels of Figure 3).

Concerning CPF exposure, we do not possess biological samples that would permit us to analyze guts histology. Indeed, despite our substantial efforts and the additional sampling campaign, we did not succeed to capture/handle/transfer (from French Polynesia to France) these specific fish. However, we performed gut length analyses on extra frozen samples that we had conserved, and we observed, as expected, that the guts’ lengthening usually occurring at metamorphosis is delayed when fish are exposed to 5 and 30 µg.l^-1^ CPF (Figure 5). This lengthening delay is similar to the one observed with NH3 treated fish (Figure 3) and is consistent with the notion that gut lengthening is controlled by TH (as evidenced in Figure 3) as CPF exposure decreases T3 levels (see Figure 5). These results on guts length and CPF exposure are now presented in Figure 5 and are discussed as follows: “However, CPF exposure at 30 µg.l^-1^ prevents guts’ lengthening at day 2 (p-value = 0.024, Figure 5 left panel), and both exposure at 5 and 30 µg.l^-1^ prevent intestines’ lengthening at day 5 (p-values = 0.047 and 0.029 respectively, Figure 5 right panel).” and: “Consistently with our observations on NH3 treated fish, exposure to CPF impairs metamorphosis, as evidenced with the repression of guts’ lengthening (Figure 5)”.

4) Please add more discussion of the gut 16S sequencing results. What does this imply about the ecology of larval vs. adult fish, and how does it integrate with the changes in craniofacial/tooth morphology and gut structure?

We agree that the discussion following our 16S sequencing experiment results (in the Results section) was too short. We have complemented this discussion and moved it in the Discussion section: “These results highlight profound anatomical and physiological transformations of the skin and the whole digestive tract (Figure 1). […] All these changes from teeth to microbiota are consistent with *A. triostegus* change of diet after entering the reefs, conforming to previous observations (Randall 1961; McCormick 1999; Frédérich et al. 2012).” As explained above, we believe that changes in the gut bacterial community in *A. triostegus* are the consequence of the concomitant diet and environmental shifts occurring at larval recruitment.

5) Are there any changes in pigment pattern, dentition or gut morphology in goitrogen-treated fish relative to controls? These should be shown, or if there are no changes, the lack of changes should be discussed.

There is apparently some sort of misunderstanding here as we did not treat the fish with goitrogens. We preferred to use NH3, a TR antagonist, to block the TH pathway instead of goitrogens (like methimazole and KClO_3_) that block the assimilation of iodine in the thyroid follicles or the TH synthesis in those same follicles (Schmidt et al. 2012). As evidenced in Figure 2, levels of TH and expressions of *TR* are peaking at and before ref entry, so the use of goitrogens on crest captured larvae would not have been appropriate (i.e., too late). In contrast the use of NH3, a TR antagonist, was, to our knowledge, the best solution to disrupt the TH pathway in crest captured larvae. As discussed in full in our response to comment 3), we have effectively scrutinized the effect of NH3 on the pigmentation pattern, the dentition and the gut morphology. As mentioned above, we do observe a consistent effect of NH3 on gut histology and we invite the reviewers to see our reply to comment 3 for more details.

6) Are there ocean samples for the other species examined in Figure 4? It would be valuable to show the increase in TH as fish approach the crest.

Following this comment we have performed TH level measurements on *Rhinecanthus aculeatus* and *Chromis viridis* oceanic larvae (near ocean samples) that we have added in Figure 4. Unfortunately and despite our effort, we were not able to sample oceanic larvae of *Chaetodon lunula* and *Ostorynchus angustatus* in neither our previous nor novel near ocean and far ocean sampling campaigns. However, results on *R. aculeatus* and *C. viridis* confirm our model of TH peaking between oceanic and crest-captured individuals (see Figure 4). We have edited the Results section for this figure as follows: “To widen our understanding of coral reef fish metamorphosis, we investigated the TH signaling during larval recruitment in four other coral fish species (*Rhinecanthus aculeatus, Chromis viridis, Chaetodon lunula* and *Ostorhinchus angustatus*) from distant families (Near et al. 2012). In these species, both T4 and T3 levels drop between day 1 and day 8, up to 3-fold for T4 (in O. angustatus, p-value = 0.029) and up to 4-fold for T3 (in *R. aculeatus*, p-value = 0.05) (Figure 4). […] This suggests some potential species-specific variation in coral reef fish TH profiles at recruitment that would be extremely interesting to decipher in future studies on additional species and with regards to other aspects of coral reef fish ecology (e.g., diets, size and pigmentation status at recruitment).”

We would like to thank the reviewers for this excellent suggestion that strengthens the generalization of our TH-mediated metamorphosis and larval recruitment model to other coral reef fishes and opens new insights for future research in this domain.

7) Are there any phenotypic aspects of metamorphosis that are blocked with CPF treatment?

As suggested by the reviewers in their comment 3, we have analyzed the impact of CPF exposure on guts lengthening (which occurs at metamorphosis). We demonstrated that CPF diminished this lengthening, in very similar way as NH3 treatment (see Figure 3 and Figure 5). We have performed these requested analyses using supplementary frozen biological samples from our previous sampling campaign. Unfortunately, these frozen samples were not appropriate for gut histology analyzes. Also, we did not analyze the effect of CPF exposure on teeth remodeling since we did not observe any effect of T3 and NH3 treatments, as well as external slope relocation, on this teeth remodeling (as discussed in our reply to reviewers’ concern 3). Considering the lengthy process of teeth remodeling (see Figure 1) and the fact that this process is already well advanced in crest individuals, it is not surprising that hormonal treatments and external slope relocation did not alter this remodeling, and it would be very unlikely that CPF exposure acts differently. Lastly, there was no effect of CPF exposure on pigmentation (see Figure 5—figure supplement 1). We invite the editor and the reviewers to see our reply to comment 3 for more details.

We believe that the additional analyses performed on guts lengthening (see Figure 5), accompanied by the initially performed analyses on the impact of CPF on *A. triostegus* feeding behavior (Figure 5) will be convincing enough for the reviewers to acknowledge that CPF is effectively impairing some aspects of metamorphosis (dropping TH levels, inhibiting guts lengthening and impairing the feeding behavior). In addition, we can also ensure that CPF is impairing other aspect of *Acanthurus triostegus* young settlers’ ecology, as we relate in a parallel study the impact of CPF on visual lateralization during settlement in *A. triostegus* (Besson et al. 2017, Exposure to agricultural pesticide impairs visual lateralization in a

larval coral reef fish, Scientific Reports, In Press). This study demonstrates that CPF can impair and even reverse visual lateralization in *A. triostegus* during this critical recruitment step.